# Resistance to Combined Anthracycline–Taxane Chemotherapy Is Associated with Altered Metabolism and Inflammation in Breast Carcinomas

**DOI:** 10.3390/ijms25021063

**Published:** 2024-01-15

**Authors:** Otília Menyhárt, János Tibor Fekete, Balázs Győrffy

**Affiliations:** 1Oncology Biomarker Research Group, Institute of Molecular Life Sciences, Hungarian Research Network, Magyar Tudósok Körútja 2, 1117 Budapest, Hungary; menyhart.otilia@med.semmelweis-univ.hu (O.M.); jfeketet@hotmail.com (J.T.F.); 2National Laboratory for Drug Research and Development, Magyar Tudósok Körútja 2, 1117 Budapest, Hungary; 3Department of Bioinformatics, Semmelweis University, 1094 Budapest, Hungary

**Keywords:** breast cancer (BC), therapy resistance, anthracycline, taxane, inflammation, innate immune response

## Abstract

Approximately 30% of early-stage breast cancer (BC) patients experience recurrence after systemic chemotherapy; thus, understanding therapy resistance is crucial in developing more successful treatments. Here, we investigated the mechanisms underlying resistance to combined anthracycline–taxane treatment by comparing gene expression patterns with subsequent therapeutic responses. We established a cohort of 634 anthracycline–taxane-treated patients with pathological complete response (PCR) and a separate cohort of 187 patients with relapse-free survival (RFS) data, each having transcriptome-level expression data of 10,017 unique genes. Patients were categorized as responders and non-responders based on their PCR and RFS status, and the expression for each gene was compared between the two groups using a Mann–Whitney U-test. Statistical significance was set at *p* < 0.05, with fold change (FC) > 1.44. Altogether, 224 overexpressed genes were identified in the tumor samples derived from the patients without PCR; among these, the gene sets associated with xenobiotic metabolism (e.g., *CYP3A4*, *CYP2A6*) exhibited significant enrichment. The genes *ORAI3* and *BCAM* differentiated non-responders from responders with the highest AUC values (AUC > 0.75, *p* < 0.0001). We identified 51 upregulated genes in the tumor samples derived from the patients with relapse within 60 months, participating primarily in inflammation and innate immune responses (e.g., *LYN*, *LY96*, *ANXA1*). Furthermore, the amino acid transporter *SLC7A5*, distinguishing non-responders from responders, had significantly higher expression in tumors and metastases than in normal tissues (Kruskal–Wallis *p* = 8.2 × 10^−20^). The identified biomarkers underscore the significance of tumor metabolism and microenvironment in treatment resistance and can serve as a foundation for preclinical validation studies.

## 1. Introduction

Breast cancer (BC) is the most common malignancy among women and is the second most common cause of cancer death after lung cancer. In 2020, 2.3 million women were diagnosed worldwide with BC, and 685,000 patients succumbed to the disease [1]. With an incidence of 1 in 8, about 13% of women are diagnosed with BC in their lifetime in the United States, accounting for one-third of all new female malignancies [2]. Incidence rates, especially of the estrogen-positive subtype, are slightly increasing in Western countries (0.5% per year) [3]; however, mortality has steadily decreased since the 1980s, with a current 5-year survival rate of 90% when diagnosed at an early stage [4].

Treatment with neoadjuvant and adjuvant chemotherapy improves long-term survival and reduces the risk of BC recurrence. The choice of therapy depends on the patient’s general condition and co-morbidities, the presence of different risk factors, and the tumor type. For over a decade, combination therapies based on anthracyclines and taxanes have been the recommended treatment options for early-stage and high-risk BC [5]. According to the 2012 Early Breast Cancer Trialists’ Collaborative Group (EBCTCG) meta-analysis, anthracycline-based regimens produced similar or better survival outcomes compared with cyclophosphamide, methotrexate, and fluorouracil (CMF) treatment, the historical standard of care; moreover, they found that adding taxane to anthracycline-based regimens further improved BC-specific and 10-year overall survival [5]. In a recent EBCTCG meta-analysis, taxanes combined with anthracyclines reduced recurrence rates compared with taxanes without anthracyclines. The highest benefits were observed when anthracyclins were added concomitantly to docetaxel plus cyclophosphamide, compared with docetaxel plus cyclophosphamide alone, leading to a 4.2 percent reduction in 10-year BC mortality (95% CI 0.4–8.1) [6]. Adding taxanes to anthracycline also reduced the 10-year risk of recurrence from 39% to 36% and the risk of BC-related mortality from 28% to 24 [6]. The effects of taxane plus anthracycline combinations were similar for estrogen receptor-positive and receptor-negative tumors, unaffected by age, lymph node status, tumor size, or grade; however, higher cumulative dose and higher-intensity treatments were more powerful [6].

Several mechanisms of action can explain the efficacy of anthracyclines: they intercalate the DNA and stabilize the topoisomerase II complex, stopping the replication process. Furthermore, anthracyclines promote cytotoxicity through free radicals while inducing histone detachment from chromatin, deregulating the epigenome and transcriptome. Thus, anthracycline treatment causes irreversible DNA damage, oxidative stress, and apoptotic cell death [7]. In contrast, taxanes bind with high affinity to microtubules and stabilize them, preventing mitotic spindle formation in the metaphase. The delayed activation of the mitotic checkpoint can trigger apoptosis [8].

Despite significant advances in the early detection of BC and a better understanding of underlying molecular mechanisms, 20–30% of BC patients are diagnosed in an advanced stage, and up to 30% of patients with early-stage breast carcinoma will relapse [9]. Although an initial clinical response and a reduction in the tumor size are frequent, the improvement is only temporary, as drug resistance can manifest within months of treatment. Consequently, innate or acquired drug resistance causes treatment failure in up to 90% of metastatic cancers [10,11,12]. Consequently, in many patients, neither a PCR nor a survival benefit can be derived from chemotherapy.

Over the past four decades, the mechanisms of chemoresistance have been extensively studied. It may involve changes in membrane transport and increased expulsion of drugs, mediated by members of the ATP-binding cassette (ABC) superfamily, particularly P-glycoprotein and multidrug-resistance-associated proteins (e.g., MDR1); this contributes toward both anthracycline and taxane resistance [13,14,15,16], along with other mechanisms related to drug metabolism [17]. Moreover, the effectiveness of anthracycline-induced cytotoxicity can be hindered by activated DNA repair mechanisms, modifications in the activity of topoisomerase II affecting the binding affinity between anthracyclines and TOP II, changes in apoptotic signaling pathways such as the tumor suppressor protein p53, alterations in cysteine and methionine metabolism, and the presence of genetic mutations [18,19,20].

Resistance to combined anthracycline–taxane treatment remains a significant obstacle in BC management. A better understanding of the mechanisms leading to resistance may improve BC treatment strategies and ultimately enhance patient survival. There is a crucial need for biomarkers that can predict therapeutic responses and aid patient classification. In this study, we aimed to identify genes that exhibit differential expression in patient samples obtained during surgery based on their subsequent response to anthracycline and taxane treatment. By comparing gene expression profiles of tumor samples, we identified genes that were upregulated in the tumors that were not responding to chemotherapy. Our findings indicate that a lack of PCR and disease recurrence is associated with the activity of metabolic pathways and inflammatory and immune processes within the tumor and its microenvironment.

## 2. Results

### 2.1. Database for the Investigation of PCR

A database comprising 634 BC patients who received combined anthracycline–taxane treatment and had information on pathological response was constructed from the following Gene Expression Omnibus (GEO) and Array Express datasets: GSE18728, GSE25066, GSE41998, and E-TABM-43 (Figure 1A). Based on the St. Gallen Consensus classification, 30.6% of patients had triple-negative BC (TNBC), 33.1% were diagnosed with luminal A, 33.3% with luminal B, and 3% with HER2-positive BC subtype (Figure 1C). In contrast, based on gene expression, the majority of patients (n = 421, ~66%) were identified with estrogen-positive BC, and 28 patients (4.4%) had HER2-positive disease; of these, 9 patient samples (1.4%) were positive for both estrogen and HER2 receptors. A total of 294 cases were lymph-node-positive, while no lymph node involvement was present in 151 patients. Altogether, 28 patients were diagnosed with grade 1, 157 with grade 2, and 231 with grade 3 disease (Figure 1C). The mean age was 49.3 years (median 48, min 24, max 75 years) (Table 1).

Most patients (96.7%) received neoadjuvant systemic treatment, while 3.3% were treated by adjuvant therapy. For most (n = 448) cases, the taxane–anthracyclin combination was the only known treatment. A total of 124 patients received neoadjuvant doxorubicin/cyclophosphamide followed by paclitaxel, and 61 patients (9.6%) were treated by adjuvant, or neoadjuvant combination of docetaxel, doxorubicin, capecitabine, and cyclophosphamide; a single patient received a neoadjuvant docetaxel–epirubicin combination (Table 1).

Of the 634 patients treated with systemic anthracycline–taxane therapy, only 31.4% developed a PCR.

### 2.2. Genes Associated with the Lack of PCR

We compared gene expression from tumor samples between responders/non-responders and determined significantly upregulated genes in patients who did not develop a PCR (*p* < 0.05; FC > 1.44).

A total of 224 overexpressed genes were identified after *p*-value correction in tumor samples not developing PCR following an anthracycline–taxane treatment, out of which 135 genes had AUC values of 0.7 or higher (Appendix A). The top three genes with the highest AUC values (*ORAI3*, *BCAM*, and *ATP6V0A1*) have been associated with cancer development. Nevertheless, the significance of these genes in restricted therapy response remains unknown, necessitating further functional investigations (Figure 2).

### 2.3. Genes Associated with Tumor Progression

We assessed expression differences among the identified 224 genes in normal, tumor, and metastatic tissue samples using TNMplot [21] and identified genes exhibiting increasing gene expression. We found ten genes with significantly higher expression in tumors than normal samples and increasing expression in metastases (*HSPB1*, *G6PD*, *BSG*, *NAPA*, *KRT18*, *BLOC1S1*, *EMP3*, *CST3*, *RHOH*, and *NELL2*). Out of these genes, the expression of *G6PD*, *NAPA*, and *BSG* genes was the highest in tumors compared to normal tissues, with a further increase in metastatic samples and particularly high AUC values in the ROC analyses (Figure 3).

### 2.4. Gene Ontology

Gene ontology analysis was performed using the top 135 upregulated genes with the strongest association to resistance (AUC ≥ 0.7) using the DAVID bioinformatics tool. The most significantly enriched term was xenobiotic catabolism (*p* = 3.28 × 10^−4^), which included four differentially expressed genes: *CYP3A4* (AUC = 0.704, *p* < 0.0001), *CYP2A6* (AUC = 0.7, *p* < 0.0001) (Figure 4A), *CYP1A2* (AUC = 0.704, *p* < 0.0001), and *FMO4* (AUC = 0.701 *p* < 0.0001). The combined AUC score of these four genes was slightly greater than any of their individual values (AUC = 0.719, *p* < 0.0001), suggesting that the combined expression of *CYP2A6*, *CYP1A2*, *CYP3A4*, and *FMO4* may be a valuable biomarker for patient stratification (Figure 4B).

### 2.5. Database for the Investigation of RFS

The data for a total of 187 BC patients treated exclusively with anthracyclines and paclitaxel, with available information on relapse-free survival (RFS), were extracted from the following datasets in GEO: GSE19615, GSE25066, GSE31519, and GSE65194 (Figure 1B). Over 90% of patients received neoadjuvant therapy. The mean RFS was 37.3 months (median 27.5, max: 89.3 months). If RFS > 60 months (5 years), patients were assigned to be responders. Of all patients, 64.7% (121) relapsed within 60 months from diagnosis.

Of the patients, 35.3% were identified to have TNBC, 20.9% were identified to have luminal A, 39.6% were identified to have luminal B, and 4.3% were identified to have HER2-positive disease (Figure 1C). In 138 cases, lymph node involvement was identified, while 47 patients were found to be node-negative. Grade 1 disease was identified in 6, grade 2 in 59, and grade 3 in 104 cases (Figure 1C). The mean age at diagnosis was 50.6 years (median 50.4, min: 24, max: 75 years) (Table 1).

### 2.6. Genes Associated with Disease Recurrence

Altogether, 51 genes had higher expression in tumor samples from patients who had relapsed within 60 months from diagnosis. We found a moderate association between the two endpoints (PCR and RFS) (Kappa = 0.241, *p* = 0.000164), with a 7.5-fold probability of developing RFS if PCR had already been reached (OR = 7.556, CI: 2.321–24.599).

We compared the list of candidate genes across the PCR and RFS groups. Interestingly, there were no overlaps between the lists of the upregulated 224 and 51 genes associated with either PCR or RFS, respectively. These genes perform entirely differently in the PCR and the RFS datasets; a comparison of their performance can be found in Appendix A (RFS vs. PCR) and Appendix A (PCR vs. RFS).

In the RFS dataset, based on AUC values, the *LYN*, *LY96*, and *HMOX1* genes separated non-responders from responders most successfully (*LYN* AUC = 0.753, *p* = 3.5 × 10^−12^; *LY96* AUC = 0.736, *p* = 3.5 × 10^−12^ (Figure 5), and *HMOX1* AUC = 0.738, *p* = 3.5 × 10^−12^).

### 2.7. Genes Associated with Tumor Progression

We evaluated expression differences among the identified 51 genes in normal, tumor, and metastatic tissue samples using TNMplot. Notably, the expression of the *SLC7A5* gene exhibited significant differences (Kruskal–Wallis test, *p* = 8.2 × 10^−20^): elevated expression was observed in tumor samples compared to normal tissues, with a further increase in metastases relative to tumors (Figure 6).

### 2.8. Inflammatory Processes Underlying Disease Recurrence

Gene ontology analysis by DAVID on the 51 genes revealed that inflammatory processes, innate immune response, and the Toll-like receptor 4 signaling pathway were the most significantly enriched terms (FDR < 13%) (Table 2). The *LYN*, *LY96*, and *ANXA1* genes assigned to these functions had exceptionally high AUC values separating responders and non-responders well in the ROC analysis (Figure 5).

## 3. Discussion

We found that 68.6% out of the 634 patients treated with combined anthracyclin–taxane systemic therapy did not develop a PCR, and 64.7% out of 187 patients relapsed within five years after receiving the anthracyclin–paclitaxel combination, emphasizing the magnitude of therapy resistance in current BC treatment strategies. In clinical practice, PCR is a meaningful measure for assessing the efficacy of neoadjuvant treatment in patients with BC and strongly correlates with RFS and overall survival, particularly in the TNBCp and HER2-positive subtypes [22]. We found a moderate association between PCR and RFS; nevertheless, the genes associated with the two endpoints did not overlap. Notably, the treatment regimens administered in the two investigated patient populations were somewhat different: in the RFS group, patients were treated exclusively by anthracyclin–paclitaxel combinations, hindering direct comparisons with the PCR group.

Over 97% of the 634 patients received neoadjuvant chemotherapy in the PCR dataset and over 90% in the RFS dataset; thus, in most cases, tumor tissue was exposed to chemotherapy before surgery. Hence, the observed changes in gene expression primarily reflect the initial response to therapy.

Our results identified altered metabolic activity in tumors of non-responders. Extensive research focuses on the connection between metabolic pathways and the development of malignancies. Metabolic syndrome, characterized by conditions like dyslipidemia, hypertension, and type 2 diabetes, has been identified as a risk factor for BC [23]. Tumor cells exhibit increased glucose consumption and often depend on amino acids, particularly leucine, for protein synthesis and growth via the mTORC1 pathway [24].

We observed that high expression of the *SLC7A5* gene, encoding a plasma membrane amino acid transporter, was associated with tumor recurrence. *SLC7A5* is crucial in fulfilling cancer cells’ elevated amino acid demand, particularly in TNBC, HER2-positive, and luminal B breast carcinomas [25] (the majority in our samples). Elevated *SLC7A5* expression correlates with larger tumor size, higher grade, distant metastases, and poor prognosis in estrogen-positive and HER2-positive BC subtypes [25]. High *SLC7A5* expression was associated with resistance to chemotherapy in breast and other tumor types [26,27]. In lung carcinomas, increased *SLC7A5* expression contributes to an immunosuppressive tumor microenvironment and reduces the efficacy of immunotherapeutic treatments [28]. The inhibitors of SLC7A5 currently being tested in clinical trials [26] could hold the potential to overcome drug resistance.

We also uncovered that a higher expression of the G6PD enzyme and genes encoding cytochrome P450 (CYP) enzymes involved in the breakdown of xenobiotics impacted the likelihood of achieving a PCR to anthracycline–taxane treatment. G6PD is involved in the pentose phosphate cycle, vital in generating ribose for DNA synthesis and maintaining redox balance through NADPH production. Increased G6PD expression has been linked to poor prognosis and resistance to chemotherapy in breast, colorectal, and lung carcinomas [29,30,31,32]. Inhibiting G6PD decreased cell viability, migration, and colonization ability while inducing autophagy through upregulating free radicals in BC cell cultures [32].

In our dataset, the high expression of genes participating in drug metabolisms, such as *CYP3A4*, *CYP2A6*, *CYP1A2*, and *FMO4*, negatively influenced the achievement of PCR. Cytochrome P450 (CYP)-dependent oxidases are heme-containing proteins involved in the degradation of compounds with an essential role in cellular signaling (including steroids, cholesterol, fatty acids, and eicosanoids), and detoxification of drugs [33]. Human CYP3A4, produced in high amounts in the liver and small intestine, is involved in the metabolism of most (~60%) of therapeutic drugs employed in the clinic [34]. CYP enzyme activity in tumors is highly relevant to the local metabolism of therapeutic agents, and many CYP genes have already been associated with the clinical efficacy of chemotherapy drugs [35]. CYP-enzymes are diverse, and their activity is context-dependent: in vitro, CYP3A4 may accelerate tumor progression independently of oncogenes through arachidonic acid metabolism by activating the PI3K/AKT and STAT3 pathways [36] and stimulates angiogenesis through increased production of vascular endothelial growth factor (VEGF) [37]. CYP3A4 protein can be detected in 20–55% of BC tissues [35], overexpressed mainly in stromal and glandular compartments [38], and in some cases, its high expression is linked to worse outcomes [39,40].

CYP3A4 and CYP2A6 play a role in developing acquired resistance toward doxorubicin in luminal A-type BC cell lines [41], and CYP3A4 also participates in taxane metabolism [35]. Increased CYP expression in tumors may limit the intracellular concentrations of docetaxel and paclitaxel, as their metabolites show no (or limited) cytotoxic activity [42,43]. CYP enzymes are activated by the human steroid xenobiotic receptor (SXR). Paclitaxel induces CYP3A4 expression via the SXR receptor, thereby preventing its own uptake and increasing its excretion [35]. Accordingly, treatment with docetaxel in MCF7 cells caused elevated CYP3A4 protein expression, which also occurred upon doxorubicin treatment [44]. Moreover, in clinical trials, patients with low CYP3A4 expression had a significantly higher response rate to docetaxel [45]. In a separate study, patients with CYP3A4-negative tumors had responded better to docetaxel, associated with more prolonged progression-free survival compared to patients with CYP3A4-positive tumors (63.2% vs. 26.1%, *p* < 0.01) [46]. In our analysis, the combined expression of the four genes participating in xenobiotic catabolism (*CYP2A6*, *CYP1A2*, *CYP3A4*, and *FMO4*) differentiated non-responders from responders with a slightly higher AUC value than any of the separate genes, confirming that CYP enzyme expression in BC is a valuable tool for predicting tumor response to anthracycline–taxane protocols. Identifying previously established interactions between gene activity and therapy response reinforces the credibility of our approach.

Moreover, our data unveiled connections between genes whose roles in cancer initiation, progression, and therapy response remain relatively unexplored. Among these genes, namely *ORAI3*, *BCAM*, and *ATP6V0A1*, high expression levels provided the most effective differentiation between non-responders and responders in the PCR dataset, yielding AUC values ranging from 0.75 to 0.77. Notably, these genes serve entirely different functions: ORAI3, a calcium ion channel, has been associated with chemotherapy resistance due to its upregulation, which hinders the expression of the p53 tumor suppressor gene and impedes apoptosis through a calcium-dependent mechanism. [47]. ORAI3, predominantly found in estrogen-positive breast tumors, promotes tumor cell division and cell cycle progression [48,49], underscoring the potential of ORAI3 as a biomarker of worse prognosis. The *BCAM* gene, also known as the basal cell adhesion molecule, encodes a protein member of the immunoglobulin superfamily and acts as a receptor for the extracellular matrix protein laminin. The function of the *BCAM* gene in BC is not well understood, but serum BCAM levels are elevated in BC patients [50]. The *ATP6V0A1* gene encodes a subunit of the vacuolar ATPase (V-ATPase) proton pump, and its role in BC requires further research. The significance of these genes in restricted therapy response remains unclear, necessitating further exploration through additional functional investigations. A distinctive strength in our study lies in proposing previously unexplored biomarker candidates, providing avenues for future validations.

In relapsed patients, we also identified an upregulated expression of genes linked to inflammatory processes and the innate immune response. In BC, inflammation is an important prognostic marker, as inflammatory cells produce factors that stimulate vascular and extracellular matrix remodeling [51]. Moreover, anthracyclines impact the immune-environment as their anticancer activity is mediated by the adaptive antitumor immune responses [52]. Previous studies uncovered markedly different immune infiltration patterns between anthracyclin–sensitive and resistant BC populations [19]. In our dataset, genes such as *LYN*, *LY96*, *ANXA1*, and *CCR1*, linked to inflammatory processes and the innate immune response, effectively distinguished patients who experienced recurrence from responders, as indicated by their AUC values above 0.72 in the ROC analysis.

*LYN*, a member of the Src receptor kinase family, has been extensively studied in BCR-ABL+ leukemia, where its persistent activation is linked to imatinib resistance [53]. In TNBC, elevated expression of LYN has been consistently linked to epithelial-mesenchymal transition, influencing cell invasion and metastatic potential [54]. Its significance extends beyond TNBC, as a high LYNA/LYNB isoform ratio is associated with poor prognosis in all breast carcinomas [55]. LY96, also known as Myeloid Differentiation Factor 2 (MD-2), is a glycoprotein involved in the initial steps of immune defense against mycobacteria. It acts as a co-receptor of Toll-like receptor 4 (TLR4), facilitating its activation through binding to its extracellular domain, initiating the innate immune response to lipopolysaccharides (LPSs) found on the outer membrane of Gram-negative bacteria [56]. *ANXA1* gene, overexpressed in non-responders in our dataset, is also involved in TLR4 regulation and the production of IFN-β induced by TLR4 [57].

One of our study’s main limitations is that we could not conduct subtype-specific investigations due to the limited sample size. Moreover, it should be noted that our dataset does not reflect the prevalence of breast tumor subtypes: the TNBC subtype accounts for 10–15% of all breast tumors, while the proportion of HER2-positive breast carcinomas is around 15–20%. In contrast, over 30% of patients in our database belonged to the TNBC subtype, while the proportion of HER2-positive patients was only 4%.

Based on the literature data, patients diagnosed with the TNBC subtype are expected to have the best response to systemic chemotherapy (e.g., anthracycline–taxane) [58]. In our dataset, of the 187 patients, 33% with the TNBC subtype responded to treatment in the long term. This value is close to the response rate described in the literature, which, depending on treatment, ranges from 30 and 50% for PCR in this subtype [58,59]. Therefore, the non-representative subtype distribution from the available data provides insight mainly into resistance mechanisms in TNBC and estrogen-positive, especially luminal B tumors. Nevertheless, according to previous findings in the literature, many identified markers upregulated in our non-responder datasets (e.g., LYN) are relevant for a broad spectrum of breast tumors.

Our findings align with prior research emphasizing the significance of various biological processes, including metabolism and immune responses [60]. In particular, our study complements previous investigations, predominantly conducted in vitro, which explored alterations in gene expression within cell cultures following doxorubicin or taxane monotherapy [60,61]. Mouse studies have implicated the modified activity of the MAPK/ERK pathway in developing resistance to doxorubicin [62]. Notably, our current study, utilizing gene expression patterns from treated tumors, provides a more nuanced understanding of the interactions between tumor cells and their immediate microenvironment, potentially accounting for identifying additional biomarkers distinct from those reported in previous in vitro studies.

## 4. Materials and Methods

### 4.1. Database Construction

To obtain datasets suitable for analyzing the therapeutic response, we searched the GEO (http://www.pubmed.com/geo, accessed on 20 March 2023 ) and Array Express (https://www.ebi.ac.uk/biostudies/arrayexpress, accessed on 22 March 2023) databases using specific keywords, including “breast”, “cancer”, “survival”, “GPL96”, “GPL570”, and “GPL571”. We focused on three Affymetrix gene array platforms (GPL96, GPL570, and GPL571) that employ identical hybridization probes for measuring gene expression. Our criteria for selecting publications included the availability of raw microarray gene expression data and clinical information regarding treatment and response. GEO datasets containing fewer than 20 samples were omitted. Some datasets initially included more than 20 specimens, but only a limited number of patients were relevant to our study. Repeatedly published arrays were identified by searching for identical expression values; only the initial one was included in the final database.

The raw CEL files were downloaded and subjected to MAS5 normalization using the Affy Bioconductor library within the R statistical environment (http://www.r-project.org, accessed on 30 March 2023) [63]. The MAS5 algorithm is designed explicitly for microarray data. It employs background correction, probe-level summarization, and global scaling to provide reliable and comparable gene expression values while addressing background noise and systematic variations within microarray datasets. We further performed scaling normalization to minimize variation between runs, setting the average expression on each chip to a standardized value of 1000. Affymetrix gene expression microarrays often detect a single gene using multiple probe sets; consequently, obtaining an unambiguous and reliable expression estimate for a specific gene can be challenging. Therefore, we employed a JetSet filter to fit a single probe to each gene [64].

### 4.2. Clinical Data

Individuals’ clinical data were collected and validated by two researchers (B.G. and J.T.F.) to ensure the accuracy and reliability of the clinical information for each sample. The clinical data examined the association between initial gene expression in tumor samples and subsequent progression among patients treated with anthracycline–taxane combination therapy. Two endpoints were defined: the absence/presence of pathological complete response (PCR) and the length of relapse-free survival (RFS). For RFS, the cutoff value was set at 60 months, and patients with RFS > 60 months were defined as responders. Patient samples censored before 60 months were excluded from the analysis.

### 4.3. Selection of Genes Associated with Treatment Response

A database with 634 patient samples was constructed to investigate genes associated with PCR, and another database with 187 samples was created to study RFS (Figure 1A,B). Gene expression information was available from these samples for 22,277 probes, representing 10,017 unique genes after applying the JetSet filter. Expression for each gene was compared between responders and non-responders for the two endpoints (PCR, RFS) by using the Mann–Whitney U-test and receiver operator characteristics (ROC). Statistical significance was accepted at *p* < 0.05 if the fold change (FC) was >1.44. In addition, we utilized the Hochberg method for *p*-value correction [65]. Only genes with a mean expression above 500 (corresponding to 50% of the all-genes mean) among non-responders were considered to highlight genes with potential relevance for future therapeutic investigations. Analysis was performed in the R statistical environment using components of the Bioconductor software package (BiocManager version: 3.18) [66].

### 4.4. Gene Ontology

We conducted gene ontology analysis on the upregulated genes observed in non-responders using the Database for Annotation, Visualization, and Integrated Discovery (DAVID) gene ontology database version 2021 [67].

### 4.5. Genes Associated with Tumor Progression

We evaluated the extent of expression differences in the identified candidate genes that distinguish responders from non-responders across normal, tumor, and metastatic tissue samples using TNMplot (www.tnmplot.com, accessed on 1 July 2023) [21] and searched for genes with increasing gene expression. We selected genes with mean expression > 500 in tumors and FC values > 1.44 between normal and tumor samples, with a further increase in metastases. For *p*-value correction, we used a 5% false discovery rate (FDR).

## 5. Conclusions

In conclusion, our study revealed that a substantial proportion of patients (68.6%) treated with combined anthracycline–taxane systemic therapy did not achieve a PCR, and a significant portion (64.7%) experienced relapse within five years, underscoring the prevalent resistance to current BC treatment strategies. Investigating gene upregulation in non-responders revealed a critical role of the modified immune environment, inflammation, and metabolism, validating some previously identified biomarkers associated with resistance to anthracycline–taxane treatment. However, the significance of many of these genes in restricted therapy response remains unclear, warranting further in-depth functional investigations. A distinctive strength of our study lies in proposing novel biomarker candidates, opening avenues for future validations to enhance patient stratification and establish a foundation for continued research in this field.

## Figures and Tables

**Figure 1 ijms-25-01063-f001:**
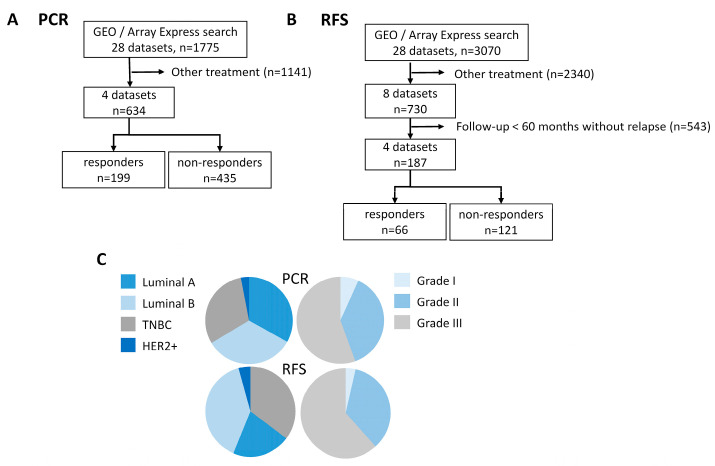
Database construction to investigate gene expression patterns related to (**A**) pathological complete response (PCR) and (**B**) recurrence-free survival (RFS). (**C**) Classification of the 634 patients in the PCR and 187 patients in RFS databases by St. Gallen Consensus subtype and tumor grade.

**Figure 2 ijms-25-01063-f002:**
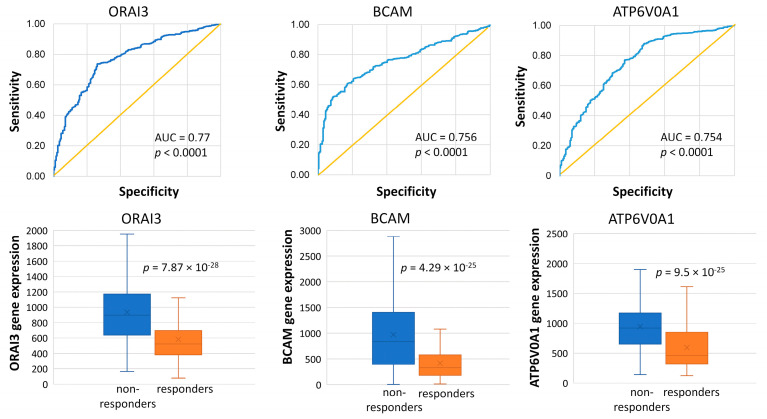
The high expression of *ORAI3*, *BCAM*, and *ATP6V0A1* differentiated non-responders from responders treated with anthracycline–taxane combination therapy in the PCR dataset (however, none of these genes differentiated responders and non-responders in the RFS dataset, as illustrated in Appendix A). The blue curve represents the discriminatory power of the model, and the yellow diagonal line serves as a baseline and represents random chance at 0.5.

**Figure 3 ijms-25-01063-f003:**
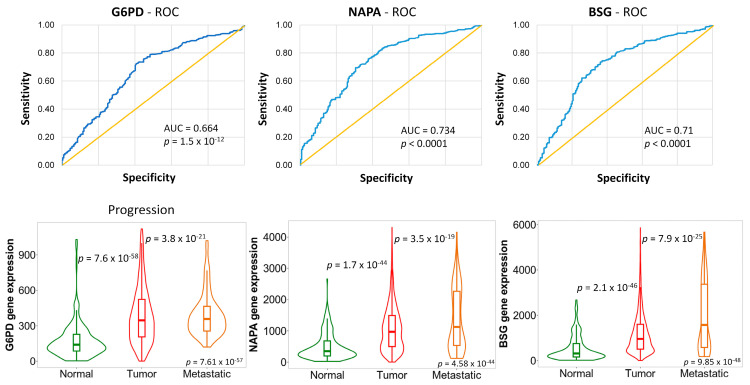
In the PCR database, high expression of *G6PD*, *NAPA*, and *BSG* genes significantly differentiated non-responders from responders. The blue curve represents the discriminatory power of the model, and the yellow diagonal line serves as a baseline and represents random chance at 0.5. *G6PD*, *NAPA*, and *BSG* genes were also associated with tumor progression, with elevated expression in tumors compared to normal samples, and a further increase in metastases. The p-values displayed in the bottom right-hand corner correspond to the Kruskal-Wallis test, assessing overall expression differences among normal, tumor, and metastatic samples. Pairwise comparisons between normal-tumor and tumor-metastatic samples were further scrutinized using the Dunn test.

**Figure 4 ijms-25-01063-f004:**
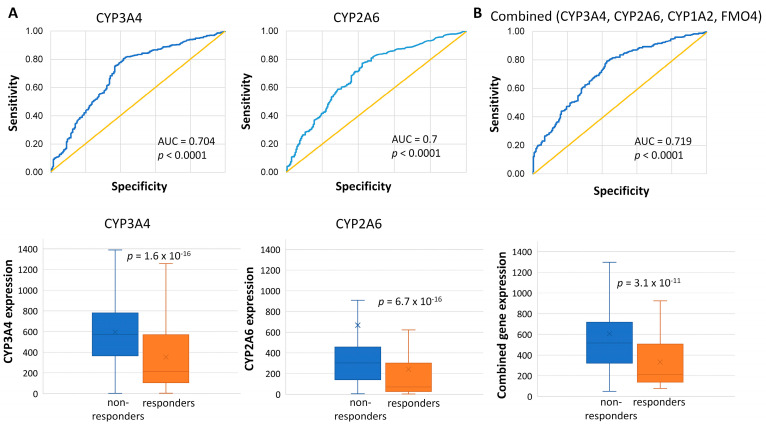
(**A**) According to DAVID gene ontology, *CYP* genes participating in xenobiotic catabolism, including *CYP3A4* and *CYP2A6*, were significantly overrepresented among the genes overexpressed in non-responders in the PCR database. (**B**) The combined AUC value of the *CYP* genes associated with xenobiotics catabolism was slightly higher than the AUC values of the individual genes.

**Figure 5 ijms-25-01063-f005:**
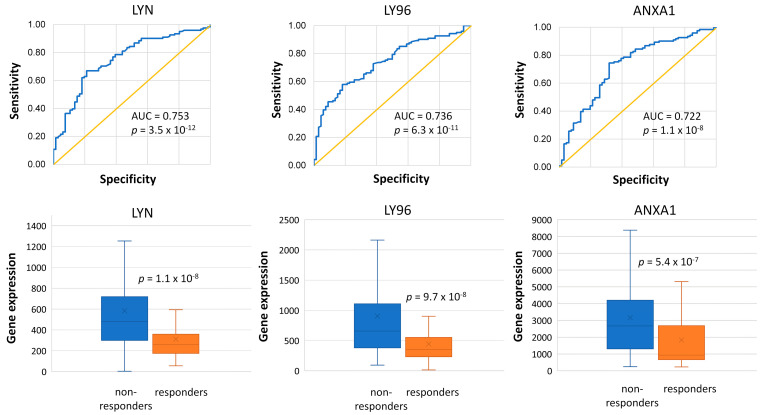
The high expression of *LYN*, *LY96*, and *ANXA1* genes were among the best that differentiated non-responders from responders treated by anthracycline-paclitaxel therapy in the RFS dataset.

**Figure 6 ijms-25-01063-f006:**
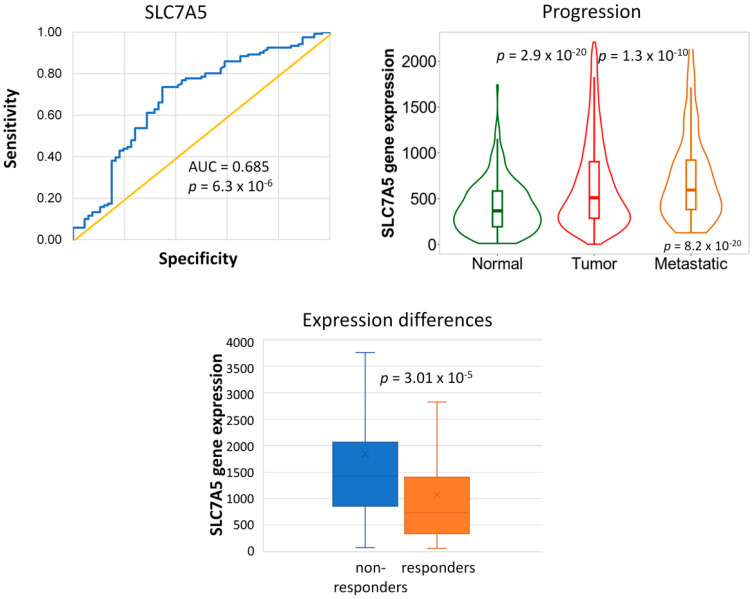
In the RFS dataset, 51 genes had higher expression in tumor samples from patients who had relapsed within 60 months from diagnosis. Among these genes, *SLC7A5* exhibited increased expression across normal samples, tumors, and metastases, as determined by TNMplot.com.

**Table 1 ijms-25-01063-t001:** Patient characteristics in the PCR and RFS datasets.

		PCR	RFS
Number of patients		634	187
Responders		199	66
BC molecular subtypes (St. Gallen)			
	Luminal A	210	39
	Luminal B	211	74
	HER2+	19	8
	TNBC	194	66
Grade			
	I	28	6
	II	157	59
	III	231	104
	unavailable	218	18
Lymph node involvement			
	yes	294	138
	no	151	47
	unknown	189	2
Age (mean)		49.3 (min 24, max 75)	50.6 (min 24, max 75)
Treatment			
	taxane and anthracycline	448	187
	docetaxel, capecitabine (adriamycin and cyclophosphamide)	61	
	doxorubicin/cyclophosphamide followed by paclitaxel	124	
	docetaxel–epirubicin	1	

**Table 2 ijms-25-01063-t002:** The genes significantly overexpressed in patients who relapsed after anthracycline-paclitaxel therapy showed enrichment of inflammatory processes and the innate immune response based on DAVID gene ontology.

GO Category	*p*-Value	Genes
Inflammatory response	4.19 × 10^−4^	*CCR1*, *AIM2*, *ANXA1*, *LY96*, *CHI3L1*, *ADM*, *CD14*
Innate immune response	5.87 × 10^−4^	*LYN*, *AIM2*, *ANXA1*, *MX2*, *SLA*, *LY96*, *DEFB1*, *CD14*
Toll-like receptor 4 signaling pathway	6.49 × 10^−4^	*LYN*, *LY96*, *CD14*

## Data Availability

The data presented in this study are available on request from the corresponding author.

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
