# Peer review of "Resistance to Combined Anthracycline–Taxane Chemotherapy Is Associated with Altered Metabolism and Inflammation in Breast Carcinomas"

_ijms, 2024, doi:10.3390/ijms25021063_

Round 1
Reviewer 1 Report
Comments and Suggestions for Authors
The manuscript entitled “Resistance to Combined Anthracycline-Taxane Chemotherapy Is Associated with Altered Metabolism and Inflammation in Breast Carcinomas” is well conceived and executed by the authors. It is completely computational work to investigate the mechanism underlying the resistance to anthracycline-taxane treatment. Authors have compared gene expression patterns with subsequent therapeutic responses.
Two cohorts were established, one for patients with PCR and the other for RFS. Overexpressed genes were identified in different cohorts and their relation to xenobiotic metabolism, inflammation, and innate immunity was observed. Also, the amino acid transporter SLC7A5 has significantly high expression in non-responders.
Overall, the manuscript is nicely written and explained systematically. The experimental design is shown properly in Figure 1, and the findings are shown in the following figures. The authors have discussed the results nicely and specifically highlighted the most important genes.
A couple of changes and additions mentioned below are required:
1. In Table 1, correct the legend. Don’t write “The 51 genes that were significantly overexpressed in patients who relapsed after….”. It looks like the authors are mentioning all 51 genes.
2. Very briefly give the rationale for the selection of the mentioned software amongst other available options for the analyses in the methods section after their first mention.
3. Add a separate section for the conclusion and future directions should be included. The findings from the study mention the altered immune environment, inflammation, and metabolism which is well-known and established in the case of tumors. Also, the role of significantly expressed genes has been known in BC or other cancers. Authors should elaborate on how their findings can specifically help and direct future research to highlight the importance of their research.
Comments on the Quality of English LanguageThe English language is fine. Minor editing is required.
Author Response
Dear Editor-in-Chief,
Thank you for the opportunity to submit a revised version of our manuscript.
Please find below our point-by-point responses to the issues raised by the reviewer (the implemented changes are also highlighted in the manuscript):
Comments from the Reviewer:
Reviewer #1:
The manuscript entitled “Resistance to Combined Anthracycline-Taxane Chemotherapy Is Associated with Altered Metabolism and Inflammation in Breast Carcinomas” is well conceived and executed by the authors. It is completely computational work to investigate the mechanism underlying the resistance to anthracycline-taxane treatment. Authors have compared gene expression patterns with subsequent therapeutic responses.
Two cohorts were established, one for patients with PCR and the other for RFS. Overexpressed genes were identified in different cohorts and their relation to xenobiotic metabolism, inflammation, and innate immunity was observed. Also, the amino acid transporter SLC7A5 has significantly high expression in non-responders.
Overall, the manuscript is nicely written and explained systematically. The experimental design is shown properly in Figure 1, and the findings are shown in the following figures. The authors have discussed the results nicely and specifically highlighted the most important genes.
Thank you for your thoughtful review of our manuscript. We appreciate the positive evaluation and would like to thank you for your time and effort in providing precious suggestions.
We have carefully considered your comments and have made the necessary revisions. Please find below our responses to your specific points and questions. We hope our revised manuscript better aligns with your expectations and the journal's standards.
A couple of changes and additions mentioned below are required:
- In Table 1, correct the legend. Don’t write “The 51 genes that were significantly overexpressed in patients who relapsed after….”. It looks like the authors are mentioning all 51 genes.
Thank you for pointing out the ambiguous wording; we have corrected the Table 1 legend per suggestion.
- Very briefly give the rationale for the selection of the mentioned software amongst other available options for the analyses in the methods section after their first mention.
Thank you for your suggestion; we have briefly elaborated on the selection of MAS5 normalization and the JetSet filter in the Methods section.
- Add a separate section for the conclusion and future directions should be included. The findings from the study mention the altered immune environment, inflammation, and metabolism which is well-known and established in the case of tumors. Also, the role of significantly expressed genes has been known in BC or other cancers. Authors should elaborate on how their findings can specifically help and direct future research to highlight the importance of their research.
Thank you for your valuable recommendations. We have included a Conclusions subsection at the end of our manuscript, where we highlight the importance of our research, please refer to page 14.

Reviewer 2 Report
Comments and Suggestions for Authors
Using publicly available datasets, Menyhart et al identified transcriptional biomarkers that can differentiate responders from non-responders of breast cancer patients receiving anthracycline-taxane chemotherapy. In addition, they have also identified gene expression markers associated with relapse free survival. Overall this is an interesting study but I still have several questions which should be easy to address.
Supplemental Table 1 is missing.
How did the authors do the quality control of the GEO datasets?
A table for characteristics of the analyzed patients is needed.
Overall, the predictive performance of genes authors highlighted should also be presented for each composed dataset. It’s unclear now that if these genes perform differently across studies.
Have the authors considered about using machine learning techniques to select genes to maximize the predictive performance?
The analyzed breast cancer patients have increased percentage of TNBC and decreased percentage of Luminal A. And these conclusion might not be generalizable to unselected breast cancer cohort. What do authors think about this?
In Figure 1, the AUCs of ORAI3, BCAM, ATP6V0A1 have a small range (0.75-0.77). Are these genes similarly important mechanistically?
In Figure 1, whats the AUC for these 3 genes on RFS?
Figure 4, the combined effect of CYP3A4 and CYP2A6 didn’t increase substantially. It seems the AUC around 0.7 is not very gene-specific.
For Figure 6, it’s unclear why authors suddenly mention SLC7A5.
What do the authors think about the clinical utility of these findings?
More detailed comments:
Figure 1. 4 dataset should be “datasets” and so on.
Page 4, line 146. “For p-value correction, we used a 5% false 146 discovery rate (FDR).” This sentence should not be in Result section.
Authors should add parts on Discussion section to discuss their transcriptional predictors of anthracycline-taxane chemotherapy and others have been reported.
Page 9, line 279,280,284 the reference format should be fixed.
Comments on the Quality of English LanguageModerate editing of English language is required.
Author Response
Dear Editor-in-Chief,
Thank you for the opportunity to submit a revised version of our manuscript.
Please find below our point-by-point responses to the issues raised by the reviewer (the implemented changes are also highlighted in the manuscript):
Comments from the Reviewer:
Reviewer #2:
Using publicly available datasets, Menyhart et al identified transcriptional biomarkers that can differentiate responders from non-responders of breast cancer patients receiving anthracycline-taxane chemotherapy. In addition, they have also identified gene expression markers associated with relapse free survival. Overall this is an interesting study but I still have several questions which should be easy to address.
We are grateful for the favorable assessment and sincerely thank you for your time and effort in offering valuable suggestions to enhance the quality of our work. Below, you will find our responses addressing your specific points and questions. We trust our revised manuscript better aligns with the journal's requirements.
Supplemental Table 1 is missing.
We apologize for the oversight; we have submitted Supplemental Table 1 along with the revised manuscript.
How did the authors do the quality control of the GEO datasets?
Thank you for the great question! GEO datasets containing fewer than 20 samples were omitted. Some datasets initially included more than 20 specimens, but only a limited number of patients were relevant to our study. Repeatedly published arrays were identified by searching for identical expression values; only the initial one was included in the final database.
We have included the above description in the revised manuscript; please refer to page 5.
A table for characteristics of the analyzed patients is needed.
Thank you for your suggestion. We have included a Table with patients' characteristics, as Table 1.
Overall, the predictive performance of genes authors highlighted should also be presented for each composed dataset. It’s unclear now that if these genes perform differently across studies.
Thank you for your valuable suggestion. We have included Supplemental Tables 2 and 3 to illustrate the gene’s performance across the two datasets; please see the clarifications on page 9.
“We compared the list of candidate genes across the PCR and RFS groups. Interestingly, there was no overlap between the lists of the upregulated 224 and 51 genes associated with either PCR or RFS, respectively. These genes perform entirely differently in the PCR and the RFS datasets; the comparison of the two gene lists can be found in Supplemental Table 2 (RFS vs PCR) and Supplemental Table 3 (PCR vs RFS).”
Have the authors considered about using machine learning techniques to select genes to maximize the predictive performance?
We have used ROC analysis, which is excellent for binary classification problems, where the class distribution is frequently imbalanced, like in our study. Nevertheless, we are planning further analysis with machine learning techniques, such as random forest, to improve the predictive performance of the candidate genes in a future study.
The analyzed breast cancer patients have increased percentage of TNBC and decreased percentage of Luminal A. And these conclusion might not be generalizable to unselected breast cancer cohort. What do authors think about this?
Thank you for the excellent point! In the revised manuscript, we discuss the subtype distribution and its consequences on page 13:
“Moreover, it should be noted that our dataset does not reflect the prevalence of breast tumor subtypes: the TNBC subtype accounts for 10-15% of all breast tumors, while the proportion of HER2-positive breast carcinomas is around 15-20%. In contrast, over 30% of patients in our database belonged to the TNBC subtype, while the proportion of HER2-positive patients was only 4%.
Based on literature data, patients diagnosed with the TNBC subtype are expected to have the best response to systemic chemotherapy (e.g., anthracycline-taxane) (62). In our dataset, of the 187 patients, 33% with the TNBC subtype responded to treatment in the long term. This is close to the response rate described in the literature, which ranges from 30-50% for pathological complete response in this subtype, depending on treatment (62, 63). Therefore, the non-representative subtype distribution from the available data provides insight mainly into resistance mechanisms in TNBC and estrogen-positive, especially luminal B tumors. Nevertheless, according to previous findings in the literature, most identified markers upregulated in our non-responder datasets are relevant for a broad spectrum of breast tumors.”
In Figure 1, the AUCs of ORAI3, BCAM, ATP6V0A1 have a small range (0.75-0.77). Are these genes similarly important mechanistically?
Thank you for the excellent observation! These genes' functions are entirely different: ORAI3 is involved in cancer progression and metastasis by encoding a functional store-operated calcium entry channel. ORAI3 has been reported to be highly expressed in different types of cancer, including breast cancer.
The BCAM gene, also known as the basal cell adhesion molecule, encodes a protein member of the immunoglobulin superfamily and acts as a receptor for the extracellular matrix protein laminin. The function of the BCAM gene, or basal cell adhesion molecule, in breast cancer is not well understood. However, it has been shown to promote the metastasis of ovarian cancer and has a putative role in cancer.
The ATP6V0A1 gene encodes a subunit of the vacuolar ATPase (V-ATPase) proton pump. Its role in breast cancer is not well understood.
Although the AUC values across these genes have a small range, the importance of these genes in limited therapy response is unknown, requiring additional functional investigations. The particular strength of our study is to suggest yet unexplored biomarker candidates for future validations.
We are discussing this issue on page 12.
In Figure 1, whats the AUC for these 3 genes on RFS?
The AUC values are:
0.527 and NOT significant for ORAI3,
0.645 for BCAM, but in the reverse direction (higher expression in responders compared to non-responders)
0.501 and NOT significant for ATP6V0A1
All these comparisons can be assessed now in Supplemental Table 3.
Figure 4, the combined effect of CYP3A4 and CYP2A6 didn’t increase substantially. It seems the AUC around 0.7 is not very gene-specific.
Thank you for your comment. AUC values may be interpreted differently based on the context. AUC values between 0.6-0.7 generally provide fair discrimination. AUC between 0.7-0.8 provides good discrimination, and between 0.8-0.9 provides very good discrimination. The discriminative power of a group of related genes close to or above 0.7 suggests their promising role in further investigations as markers in patient stratification.
Also, as discussed in the manuscript, many CYP genes have already been associated with the clinical efficacy of chemotherapy drugs. Identifying previously established interactions between gene activity and therapy response reinforces the credibility of our approach (please see page 12).
For Figure 6, it’s unclear why authors suddenly mention SLC7A5.
Thank you for your observation; we have corrected the Figure legend to increase the clarity of the manuscript.
“In the RFS dataset, 51 genes had higher expression in tumor samples from patients who had relapsed within 60 months from diagnosis. Out of these genes, the expression of the SLC7A5 increased significantly across normal samples, tumors, and metastases.”
What do the authors think about the clinical utility of these findings?
Thank you for your question. In this revised manuscript, we elaborate on the clinical utility of our findings; please refer to page 14 of the manuscript and the Conclusions section.
“Notably, our current study, utilizing gene expression patterns from treated tumors, provides a more nuanced understanding of the interactions between tumor cells and their immediate microenvironment, potentially accounting for identifying additional biomarkers distinct from those reported in previous in vitro studies.”
“Investigating gene upregulation in non-responders revealed a critical role of the modified immune environment, inflammation, and metabolism, validating some previously identified biomarkers associated with resistance to anthracycline-taxane treatment. However, the significance of many of these genes in restricted therapy response remains unclear, warranting further in-depth functional investigations.
A distinctive strength of our study lies in proposing novel biomarker candidates, opening avenues for future validations to enhance patient stratification and establish a foundation for continued research in this field.”
More detailed comments:
Figure 1. 4 dataset should be “datasets” and so on.
Thank you for your observation; we have corrected Figure 1. We have also revised the manuscript for English language accuracy; all modifications are highlighted.
Page 4, line 146. “For p-value correction, we used a 5% false 146 discovery rate (FDR).” This sentence should not be in Result section.
Thank you for your observation; we have included the description and particulars of gene selection in the Methods section instead.
Authors should add parts on Discussion section to discuss their transcriptional predictors of anthracycline-taxane chemotherapy and others have been reported.
Thank you for your suggestion. We have included a paragraph about how our study fits into the context of previous findings; please refer to page 13:
“Our findings align with prior research emphasizing the significance of various biological processes, including metabolism and immune responses (64). In particular, our study complements previous investigations, predominantly conducted in vitro, which explored alterations in gene expression within cell cultures following doxorubicin or taxane monotherapy (64, 65). Mouse studies have implicated the modified activity of the MAPK/ERK pathway in developing resistance to doxorubicin (66). Our current study, utilizing gene expression patterns from treated tumors, provides a more nuanced understanding of the interactions between tumor cells and their immediate microenvironment, potentially accounting for identifying additional biomarkers distinct from those reported in previous in vitro studies.”
Page 9, line 279,280,284 the reference format should be fixed.
Thank you for your observation; we have fixed the format of the requested references.
We trust that the concerns raised regarding the manuscript have been adequately addressed in this revised version, as our commitment is to deliver a well-informed and balanced document. We express our gratitude once more for the valuable input of the two reviewers, which unquestionably enhanced the quality of our work.
Sincerely,
Otília Menyhart, János Tibor Fekete and Balázs GyÅ‘rffy
Round 2
Reviewer 2 Report
Comments and Suggestions for Authors
The authors have sufficieent response to the review questions.
Comments on the Quality of English LanguageMinor editing of English language required